# The Role of CX3CL1 and ADAM17 in Pathogenesis of Diffuse Parenchymal Lung Diseases

**DOI:** 10.3390/diagnostics11061074

**Published:** 2021-06-11

**Authors:** Jan Urban, Magda Suchankova, Martina Ganovska, Vladimir Leksa, Frantisek Sandor, Eva Tedlova, Brian Konig, Maria Bucova

**Affiliations:** 14th Department of Pneumology and Phthisiology, National Institute for Tuberculosis, Respiratory Diseases and Thoracic Surgery, 059 84 Vysne Hagy, Slovakia; 2Institute of Immunology, Faculty of Medicine Comenius University, 811 08 Bratislava, Slovakia; magda.suchankova@gmail.com (M.S.); maria.bucova@fmed.uniba.sk (M.B.); 3Department of Clinical Laboratories, National Institute for Tuberculosis, Respiratory Diseases and Thoracic Surgery, 059 84 Vysne Hagy, Slovakia; ganovskamartina@gmail.com; 4Institute of Molecular Biology, Slovak Academy of Sciences, 845 51 Bratislava, Slovakia; vladmir.leksa@savba.sk; 5Centre for Pathophysiology, Infectiology and Immunology, Institute for Hygiene and Applied Immunology, Medical University of Vienna, A-9010 Vienna, Austria; 6Department of Pneumology and Phthisiology, Faculty of Medicine Comenius University and University Hospital, 821 01 Bratislava, Slovakia; frantisek.sandor@gmail.com (F.S.); evatedlova@hotmail.com (E.T.); 7Department of Operations Research and Econometrics, Faculty of Economic Informatics, University of Economics in Bratislava, 852 35 Bratislava, Slovakia; konigbrian@gmail.com; 8Institute of Economic Research of Slovak Academy of Sciences, 811 05 Bratislava, Slovakia

**Keywords:** CX3CL1, ADAM17, diffuse parenchymal lung diseases, fibrotic process, bronchoalveolar lavage fluid

## Abstract

Fractalkine (CX3CL1) is a unique chemokine that functions as a chemoattractant for effector cytotoxic lymphocytes and macrophages expressing fractalkine receptor CX3CR1. CX3CL1 exists in two forms—a soluble and a membrane-bound form. The soluble CX3CL1 is released from cell membranes by proteolysis by the TNF-α-converting enzyme/disintegrin-like metalloproteinase 17 (TACE/ADAM17) and ADAM10. In this study, we evaluated the diagnostic relevance and potential roles of CX3CL1 and ADAM17 in the pathogenesis of diffuse parenchymal lung diseases (DPLDs) in the human population. The concentration of CX3CL1 and ADAM17 was measured by the enzyme-linked immunosorbent assay (ELISA) test in bronchoalveolar lavage fluids of patients suffering from different DPLDs. The concentration of CX3CL1 was significantly higher in patients suffering from idiopathic pulmonary fibrosis (IPF) and hypersensitivity pneumonitis patients compared to the control group. A significantly higher concentration of CX3CL1 was measured in fibrotic DPLDs compared to non-fibrotic DLPD patients. We found a positive correlation of CX3CL1 levels with the number of CD8+ T cells, and a negative correlation with CD4+ T cells in BALF and diffusion capacity for carbon monoxide. The concentration of ADAM17 was significantly lower in the IPF group compared to the other DPLD groups. We noticed a significantly higher CX3CL1/ADAM17 ratio in the IPF group compared to the other DPLD groups. We suggest that CX3CL1 has a distinctive role in the pathogenesis of DPLDs. The level of CX3CL1 strongly correlates with the severity of lung parenchyma impairment. The results suggest that high values of CX3CL1/ADAM17 could be diagnostic markers for IPF.

## 1. Introduction

Diffuse parenchymal lung diseases (DPLDs) comprise a broad heterogeneous group of diseases with more than 200 various diagnostic units. The common feature of DPLDs is the involvement of the lung interstitium characterized by different degrees of inflammation and fibrosis with extracellular matrix accumulation [1]. The prognosis depends on the underlying immunopathological process, which is in the vast majority of cases an inflammatory response to a known or unknown trigger. The continuing inflammatory process can result in the activation of the fibrosing processes [2]. The worst prognosis is related to the progressive fibrotic phenotype of the disease [3]. The main problem of DPLDs in regard to the number of diagnostic units is the differential diagnosis and also the limited treatment possibilities. The cornerstone to move ahead in therapeutic options is an improvement of comprehension of the immunopathogenesis. 

Fractalkine (CX3CL1) is the only member of the CX3C-chemokine family [4]. CX3CL1 occurs in two forms—a membrane-bound form and a soluble form that acts as a strong chemoattractant and supports migration of natural killer (NK) cells, cytotoxic T cells and macrophages [5]. The membrane-bound form is expressed on endothelial cells as an adhesion molecule, and it supports leukocyte migration to the site of inflammation. Its expression increases under the influence of proinflammatory cytokines—tumor necrosis factor (TNF)-α, interleukin (IL)-1 and interferon (IFN)-γ [6,7]. CX3CL1 is also expressed by epithelial cells in the lungs, kidneys and intestines [8,9]. The soluble CX3CL1 is released from cell membranes by proteolysis by TNF-α-converting enzyme/disintegrin-like metalloproteinase 17 (TACE/ADAM17) and ADAM10 [10,11]. CX3CL1 can bind to its specific receptor (CX3CR1) in a 1:1 ratio [12]. CX3CR1 is expressed by the majority of CD16+ NK cells, CD14+ monocytes and the majority of CD3+ T cells. CX3CR1 is also a membrane marker of cytotoxic NK and T cells [13,14].

On the basis of the growing amount of evidence, the contribution of the CX3CL1-CX3CR1 axis has been assumed to contribute to the pathogenesis of vascular damage and inflammatory diseases, such atherosclerosis [15], rheumatoid arthritis [16], bronchial asthma [17] and emphysema [18]. 

The role of CX3CL1 in the pathogenesis of DPLDs has recently been demonstrated in Ishida’s murine research work [19] and in several studies in humans [20,21].

In this study, we evaluated the potential role of CX3CL1 in the pathogenesis of the fibrotic process of various DPLDs. The aim of this study was to determine the level of CX3CL1 in the bronchoalveolar lavage fluid (BALF) of patients affected by particular DPLDs—pulmonary sarcoidosis (PS), idiopathic pulmonary fibrosis (IPF), hypersensitivity pneumonitis (HP) and connective tissue disease-associated interstitial lung disease (CTD-ILD). We compared concentrations of CX3CL1 in each of these populations, in particular fibrotic and non-fibrotic stages of DPLDs. Next, we correlated the level of CX3CL1 with the number of cells in BALF and the severity of lung impairment evaluated by measurement of lung diffusion capacity for carbon monoxide (DLCO). In the light of previously published data regarding ADAM17 [10,11] acting as an enzyme releasing a soluble form of CX3CL1, we also evaluated the relationship of CX3CL1 to ADAM17 concentrations in BALF. Finally, we statistically evaluated the ability to distinguish the primary fibrotic process (IPF) from secondary fibrotic (fibrotic stages of HP, PS, CTD-ILD) and non-fibrotic DPLDs on the basis of the levels of studied molecules in BALF.

## 2. Materials and Methods

### 2.1. Study Group

The study group comprised 270 DPLD patients divided into 4 main groups. We had 156 patients with pulmonary sarcoidosis (PS), 46 patients with idiopathic pulmonary fibrosis (IPF), 46 patients with hypersensitivity pneumonitis (HP) and 22 patients with connective tissue disease-associated interstitial lung disease (CTD-ILD). The diagnosis of PS, IPF, HP and CTD-ILD was established in accordance with current expert opinions [22], clinical guidelines [23,24] and previously published evidence [25,26] as the result of multidisciplinary team consensus (pneumologist, radiologist and pathologist) in a tertiary health care center specialized in pulmonary medicine. Subsequently, two subgroups—fibrotic group and non-fibrotic group—were created from the PS, HP and CTD-ILD groups of patients according to the presence of fibrotic changes in high-resolution computed tomography (HRCT) scans.

The fibrotic group labelled as OFI (other fibrosis than IPF) comprised 46 patients—7 subjects from the PS group (stage IV. of sarcoidosis), 21 subjects from the HP group (chronic/fibrotic HP) and 18 subjects from the CTD-ILD group (fibrotic stage of CTD-ILD).

The non-fibrotic group labelled in our study as NFI (non-fibrosis) comprised 178 patients—149 subjects from the PS group (stage I.-III. of sarcoidosis), 25 subjects from the HP group (inflammatory/non-fibrotic HP) and 4 subjects from the CTD-ILD group. The control group (CG) included 16 subjects without clinical and radiological signs of DPLD, and without a considerable pathology in the BALF differential cell count. All subjects included in the study had no other respiratory comorbidities and no history of active malignant disease. All patients included in our study were newly diagnosed with stable disease. All patients out of the IPF, HP and PS study groups were at the time of BALF sampling without any antifibrotic and immunosuppressive treatment. Patients from the CTD-ILD group were treated by immunosuppressive agents at the time of establishing the diagnosis of CTD-ILD. This medication had previously been prescribed by a rheumatologist.

### 2.2. Procedures and Sample Processing

BALF samples were obtained from all study subjects during the bronchoscopy in complying with international guidelines [27]. The lung segment was lavaged by instilling a sterile saline solution up to a maximum volume of 160 mL, in four 40 mL aliquots, with immediate aspiration between each aliquot volume. BALF was first filtered through a double layer of sterile gauze, centrifuged (300× *g* for 15 min at 10 °C), and supernatants were collected. Finally, the pelleted cells were used for flow cytometry using a NAVIOS Flow Cytometer (Beckman Coulter France S.A.S, Villepinte, France). The numbers of lymphocytes and lymphocyte subsets were determined using the tetraCHROME CD45-FITC/CD4-PE/CD8- ECD/CD3-PC5 Antibody Cocktail (Beckman Coulter France S.A.S, Villepinte, France) (Appendix A). Data were analyzed using KALUZA software (Beckman Coulter France S.A.S). Both the percentage and absolute count of CD3, CD4 and CD8 cells were determined. The total number of cells (CD3+, CD4+, CD8+) was calculated from the absolute count of BALF cells evaluated by a flow cytometer (absolute count of cells per milliliter was obtained from filtered, not concentrated, not centrifuged BALF by Flow-Count Fluorospheres (Beckman Coulter France S.A.S, Villepinte, France). Data are presented in Table 1.

The concentration of CX3CL1 was measured by the enzyme-linked immunosorbent assay (ELISA) kit (Human Fractalkine Elisa test, FineTest, Wuhan Fine Biotech Co., China, Catalogue No. EH0141) and ADAM17 using the Human ADAM17 (disintegrin and metalloproteinase domain-containing protein 17) ELISA Kit (FineTest, Wuhan Fine Biotech Co., China, Catalogue No. EH1488) precisely according to the instructions recommended by the manufacturer.

Diffusion capacity for carbon monoxide (DLCO) measurement was performed on all subjects during the diagnostic process by using the single breath method in accordance with 2017 ATS/ERS standards [28].

### 2.3. Statistical Analysis 

The one-sample Kolmogorov–Smirnov test was used to determine whether the investigated population followed a normal distribution. Mann–Whitney or non-parametric analysis of variance (Kruskal–Wallis) with the Dunn post-test was used to determine the differences and the statistical significance. The results were expressed as the median and interquartile range (IQR). Correlation analysis was performed by Spearman test. Logistic regression models (logit) were used to verify the effect of particular biomarkers on binary defined lung diagnosis variables after controlling patient characteristics. We considered a *p*-value < 0.05 indicating the statistical significance. In order to discriminate fibrotic lung diagnosis from other diseases through the analyzed biomarkers, we adopted ROC curves. Presented ROC curves plot sensitivity and specificity levels calculated from logit models. We published sensitivity and specificity levels achieved at the highest values of Youden’s index. Statistical analysis was performed using SAS and Stata software.

## 3. Results

### 3.1. Increased Concentration of CX3CL1 in Patients with IPF and HP

Compared to the control group (CG), patients with IPF and HP had significantly higher levels of CX3CL1 in BALF (*p* < 0.01 and *p* < 0.05, respectively). In the PS and CTD-ILD groups, we did not observe a significant difference compared to CG. In comparison among different DPLD groups, we found significantly lower CX3CL1 concentrations in the PS group (*p* < 0.001) compared to all other DPLD groups (IPF, HP, CTD-ILD). No difference was observed in CX3CL1 concentration among the IPF, HP and CTD-ILD groups (Figure 1a). We found significant differences in CX3CL1 concentration in both fibrotic groups, IPF and OFI (other fibrosis than IPF) groups, compared to the non-fibrotic (NFI) group of patients (*p* < 0.001). No significant difference was observed between the IPF and OFI groups (Figure 1b). 

In order to evaluate the statistical significance of the CX3CL1 concentration difference between fibrotic pulmonary diseases (IPF and OFI) and non-fibrotic lung diseases (NFI) or the control group, we adopted logistic regression models (Table 2). The elevated levels of CX3CL1 in BALF correlate highly significantly with the development of fibrosis and raised levels of CX3CL1 are associated with an increased probability of the fibrotic lung process. CX3CL1 data remained highly significant even after controlling of other specific parameters (age, sex and smoking). 

Based on the high statistical significance of CX3CL1, we decided to test a possible predictive ability of CX3CL1 to discriminate fibrotic from non-fibrotic DPLDs. The ROC curve and the AUC (area under curve) value (0.8182) show that CX3CL1 as a biomarker has a high predictive ability to discriminate fibrotic lung processes from other non-fibrotic DPLD diagnoses (Figure 2). The highest level of Youden’s index (0.5586), which provides information on diagnostic test performance, was achieved at the values of sensitivity = 0.7802 and specificity = 0.7784.

### 3.2. Positive Correlation of the Level of CX3CL1 with Number of Macrophages, Neutrophils and CD8+ T Cells, and a Negative Correlation with the Number of CD4+ T Cells and with DLCO Values 

Correlation analysis of CX3CL1 concentration with a differential cell count in BALF in all study groups pointed to a positive correlation with absolute BALF cell count, absolute count of neutrophils, macrophages, CD8+ (cytotoxic) T cells, and a negative correlation with an absolute count of CD4+ (helper) T cells, CD4+/CD8+ T cell ratio. Moreover, the CX3CL1 concentration inversely correlated with DLCO levels (Table 3).

### 3.3. Soluble ADAM-17 Concentrations in BALF Are Elevated in HP and Decreased in IPF Patients

ADAM17 is an enzyme that cleaves membrane-bound CX3CL1 from the cell surface by proteolysis and releases the soluble CX3CL1 form. We evaluated soluble ADAM17 concentration in BALF in different study groups (Figure 3a). We found significantly higher levels of ADAM17 in the BALF (*p* < 0.01) of HP patients compared to CG. We did not find a significant difference in the levels of ADAM17 in any other DPLD groups compared to CG. Furthermore, we found significantly lower ADAM17 concentrations in the IPF group compared to the HP (*p* < 0.001), PS and CTD-ILD groups (*p* < 0.05). Subsequently, comparing the ADAM17 concentrations among the IPF, OFI and NFI groups, we found significantly lower concentrations of this molecule in the IPF group compared to both the OFI and NFI groups (*p* < 0.01). No significant difference in ADAM17 concentration was detected between the OFI and NFI groups (Figure 3b).

### 3.4. Positive Correlation of the Level of ADAM17 with Absolute Cell Count in BALF

Correlation analysis of ADAM17 concentration with differential cell counts in BALF in all study groups pointed to a positive correlation with the absolute BALF cell count, absolute count of lymphocytes (both CD4+ T cells and CD8+ T cells) and macrophages (Table 4). 

### 3.5. IPF Patients Had the Highest CX3CL1/ADAM17 Ratio

Finally, we evaluated the CX3CL1/ADAM17 ratio (Figure 4a) and found a significantly higher value of the CX3CL1/ADAM17 ratio in IPF compared to CG (*p* < 0.01). The significant difference in the value of this ratio was found in IPF compared to all the other DPLD groups—PS, HP and CTD-ILD (*p* < 0.001, *p* < 0.01 and *p* < 0.01, respectively) as well. No other significant differences were found in comparison among other DPLD groups to each other and CG as well. Furthermore, we also found a significant difference in CX3CL1/ADAM17 values between the IPF group and both the OFI and NFI subgroups (*p* < 0.01 and *p* < 0.001, respectively). A significant difference was also found between the OFI and NFI groups (*p* < 0.01) (Figure 4b).

The assumption that a higher value of CX3CL1/ADAM17 ratio is associated with a higher probability of having IPF was further supported by logistic regressions (Table 5). After correcting for other variables that may affect the development IPF (age, sex and smoking), the impact of CX3CL1/ADAM17 ratio remained positively statistically significant, meaning that a higher ratio of CX3CL1/ADAM17 is associated with a higher probability of having IPF.

Based on our results demonstrating the high CX3CL1/ADAM17 value in IPF patients, we used logistic regression analysis to evaluate the diagnostic ability of CX3CL1/ADAM17 testing to discriminate IPF from other DPLD diagnoses. The ROC curve and the AUC value (0.8424) show that CX3CL1/ADAM17 ratio as a biomarker has a high predictive value to discriminate IPF from other DPLD diagnoses (Figure 5). The highest level of Youden’s index (0.6657) was achieved at the values of sensitivity = 0.8077 and specificity = 0.858.

Finally, we were interested in answering the question whether it was possible to discriminate IPF from OFI by means of CX3CL1/ADAM17 ratio. The ROC curve and the AUC value (0.7575) show that CX3CL1/ADAM17 ratio as a biomarker has a moderate predictive ability to discriminate IPF from OFI (Figure 6). The highest level of Youden’s index (0.4936) was achieved at the values of sensitivity = 0.5769 and specificity = 0.9167.

## 4. Discussion

The participation of the CX3CL1-CX3CR1 system in the pathogenesis of DPLDs had not been described until recently, when Ishida et al. reported for the first time the possible role of this system in the pathogenesis of lung fibrosis [19]. In a murine experimental model of bleomycin-induced lung fibrosis they demonstrated a significantly lower degree of fibrosis in CX3CR1-deficient mice compared to wild-type mice. Furthermore, this work demonstrated higher amounts of M2 (profibrotic)-macrophages in the BALF of CX3CR1-expressing mice in comparison to M1-macrophages, which was in contrast to an opposite ratio of macrophage types in CX3CR1-deficient mice. Moreover, lung tissue of CX3CR1-deficient mice contained lower amounts of fibrocytes in comparison to CX3CR1-expressing mice. Finally, lower levels of TGF (transforming growth factor)-β were measured in the lungs of CX3CR1-deficient mice (gene and protein level as well) compared to CX3CR1-expressing mice.

According to this study, a plausible role of the CX3CL1-CX3CR1 system has been hypothesized in the intrapulmonary recruitment of profibrotic (M2) macrophages and fibrocytes, which would contribute to the development of bleomycin-induced fibrosis by TGF-β production [19].

Only a few studies reporting the role of the CX3CL1-CX3CR1 system in the pathogenesis of DPLDs in humans have been published so far. Hoffman-Vold et al. [20] compared the concentration of CX3CL1 in the serum and lung tissue of patients affected by systemic sclerosis-associated interstitial lung disease (SSc-ILD) and in healthy controls. The CX3CL1 concentration was higher in serum and lung tissue as well, when compared to a healthy control group and significantly inversely correlated with DLCO, positively correlated with the concentration of anti-topoisomerase-I-antibody and progression of lung involvement. Regarding this study, CX3CL1 appears to be associated with progressive lung involvement in systemic sclerosis.

Recently, Greiffo et al. [21] focused on the role of CX3CL1 in DPLDs. This study reported the importance of CX3CL1 acting for transmigration of CX3CR1-positive non-classical monocytes from the blood into the human fibrotic lung in different DPLDs (HP, CTD-ILD and non-specific interstitial pneumonia). This study also demonstrated that bronchoepithelial cells were the major source of CX3CL1 in DPLDs.

It is well known that IPF is primarily a fibrotic disease with a very low degree of inflammation [29]. Chronic HP, CTD-ILD and PS can also develop the potential of the fibrotic process [30]. In accordance with previous studies, in our study we have demonstrated that CX3CL1 is implicated in the fibrotic process in IPF and OFI (stage IV. PS, fibrotic HP and CTD-ILD). Furthermore, we have shown a significantly higher level of CX3CL1 in BALF in the IPF and OFI groups of patients compared to the CG group. No difference was found in CX3CL1 concentration between the IPF and OFI subgroups; however, a statistically significant difference was found between both these two fibrotic groups and the NFI subgroup. These results underline the current evidence of CX3CL1 being an important element in fibrotic process development. The potential of CX3CL1 in the pathogenesis of fibrosing lung disease development was shown to be valid regardless of the origin of fibrosis—be it a primary (IPF) or secondary fibrotic process (HP, CTD-ILD, PS). Subsequently, the lowest CX3CL1 concentration in the PS study group may be related to the lowest fibrotic potential of PS associated with specific inflammatory response (less contribution of CD8+ T cells) compared to HP and CTD-ILD [31,32].

Our results point to a strong negative correlation of CX3CL1 concentration with DLCO, which indicates an obvious association with the severity of lung parenchymal damage. This observation is similar in two previous studies performed in humans [20,21].

Negative and positive correlations of CX3CL1 concentration in our study with percentage and absolute count of CD4+ (helper) T cells and CD8+ (cytotoxic) T cells, respectively, are in accordance with previous evidence about CX3CL1′s role as a chemoattractant for cytotoxic (CX3CR1-expressing) cells [13]. As mentioned above, CD8+ T cells are assumed to play a role in the pathogenesis of pulmonary fibrosis [31,32]. In line with our results, we assume that the higher the CX3CL1 concentration is in the lung parenchyma, the more it attracts cytotoxic cells including CD8+ T cells, thus enhancing the fibrotic milieu. The assumed abundant source of CX3CL1 comprises inflammation damaged bronchoepithelial cells [21]. Nowadays, it is not clear if CX3CL1 is a causal factor in the fibrotic process (not only in IPF) or only the consequence of other mechanisms acting in the pathogenesis of the fibrotic process. There is a need for continuing research.

From previous research studies it is known that membrane-bound CX3CL1 is released from the cell surface by proteolytic cleavage by ADAM10 and ADAM17 [10,11]. Our results support the hypothesis that ADAM17 plays a key role in the shedding of soluble CX3CL1 in HP and CTD-ILD; however, in IPF probably another mechanism acts in CX3CL1 shedding. Based on this study, it is not yet possible to identify the mechanism that leads to the release of CX3CL1 in IPF. Some studies demonstrated the role of ADAM17 in advancing the inflammatory response, and the lack of ADAM17 was shown to result in a decrease of leukocyte migration into the inflammation site [33,34,35]. Low ADAM17 concentration in the IPF group in our study correlates with previous studies analyzing the role of ADAM17 in inflammatory processes. It could explain the low degree of ongoing inflammation in IPF compared to other DPLD groups. Positive correlation of ADAM17 concentration in our study with the abundance of immune cells in BALF (total cell counts, absolute number of macrophages, neutrophils and lymphocytes) can serve as a potential marker of low ongoing degree of inflammation. This also correlates with the role of ADAM17 in the inflammatory process. Following the current evidence for the role of ADAM17 in the inflammatory process and its positive correlation with immune cell count in our study, we could regard HP as the DPLD with the highest degree of inflammation. The mechanisms leading to low ADAM17 concentration in IPF are currently unknown and need to be explored by further research. One of the possible factors leading to low ADAM17 activity in IPF could be the tissue inhibitor of metalloproteinase-3 (TIMP-3), the natural inhibitor of this enzyme [36]. The study of García-Alvarez et al. demonstrated significantly increased TIMP-3 gene expression and its protein presence localized in the fibroblastic foci and the extracellular matrix in IPF tissues [37]. 

The high value of the CX3CL1/ADAM17 ratio in the IPF group of patients compared with other DPLD groups is also very interesting. A high value of AUC points out the possibility to construct a useful test that could potentially provide high sensitivity and specificity with a good ability to differentiate IPF from other DPLD diagnoses.

Finally, we demonstrated the high CX3CL1/ADAM17 ratio in the IPF group with a significant difference in comparison to a subgroup of subjects with other fibrotic DPLDs (OFI group). Depending on this significant difference, CX3CL1/ADAM17 ratio could be considered a helpful diagnostic marker distinguishing between IPF and fibrotic stages of other DPLDs. We fully realize that there is a further need for a higher number of subjects and the necessity of future studies to support and validate the CX3CL1/ADAM17 ratio as an appropriate biomarker. This marker could potentially serve as a useful prognosis assessment tool in clinicians’ armamentarium to differentiate the diagnosis of IPF to other progressive fibrosing DPLDs. 

## 5. Conclusions

Our study demonstrates that CX3CL1 is involved in the pathogenesis of DPLDs. Its concentration in BALF strongly positively correlates with the severity of lung parenchyma impairment. Correlation analysis revealed that CX3CL1 might influence the activity of cytotoxic T cells. We dare to hypothesize that on the basis of higher CX3CL1 concentration in fibrotic stages compared to non-fibrotic stages of DPLDs, CX3CL1 considerably advances the fibrosing process. The soluble form of CX3CL1 is released by proteolytic cleaving by enzyme ADAM17 in the majority of DPLDs. However, according to the results of our study the mechanism of CX3CL1 shedding seems to be different in the IPF patient population. This was demonstrated by the high value of the CX3CL1/ADAM17 ratio in the IPF group only. At last, there is potential for this ratio to be used as a specific diagnostic marker for patients suspected of having the diagnosis of IPF.

## Figures and Tables

**Figure 1 diagnostics-11-01074-f001:**
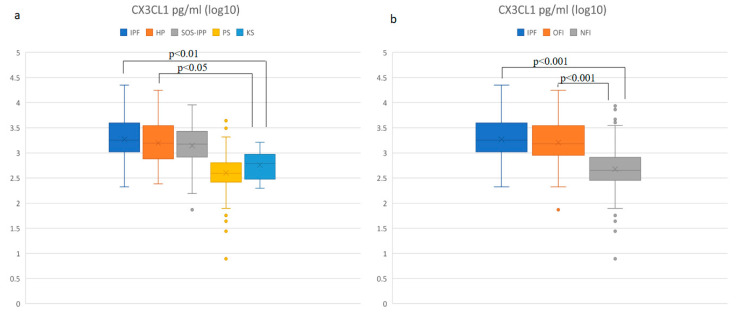
Comparison (log10) of the CX3CL1 concentration in BALF: (**a**) in DPLD groups and control group, (**b**) in idiopathic pulmonary fibrosis (IPF), other fibrosis than IPF group (OFI) and non-fibrosis group (NFI). Idiopathic pulmonary fibrosis (IPF)—median: 1803.3 pg/mL, interquartile range (IQR): 2668.7; hypersensitivity pneumonitis (HP)—median: 1566.0 pg/mL, IQR: 2708.2; connective tissue disease-associated interstitial lung disease (CTD-ILD)—median: 1502.9 pg/mL, IQR: 2131.8; pulmonary sarcoidosis (PS)—median: 393.9 pg/mL, IQR: 373.6; control group (CG)—median: 616.8 pg/mL, IQR: 633.4; OFI—median: 1443.8 pg/mL, IQR: 2457.4; NFI—median: 448.7 pg/mL, IQR: 529.0.

**Figure 2 diagnostics-11-01074-f002:**
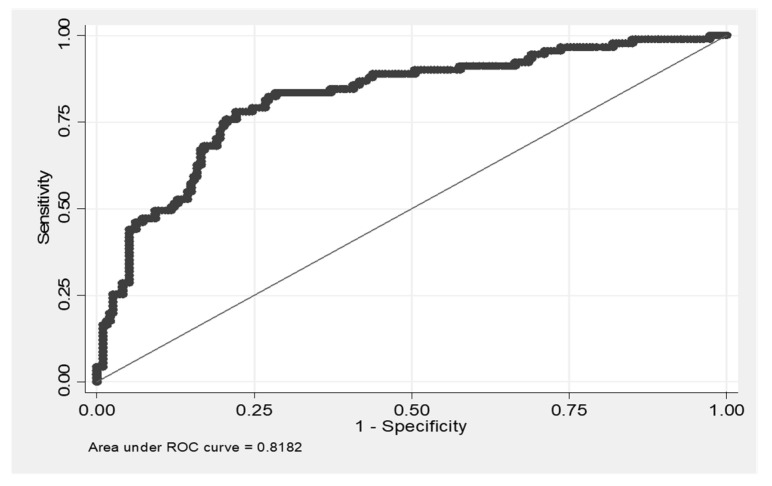
ROC curve of CX3CL1 predictive ability to discriminate fibrotic diffuse parenchymal lung disease (DPLD) from other (non-fibrotic) DPLDs. Youden’s index = 0.5586; the cut-off value of CX3CL1 associated with the highest value of Youden’s index is CX3CL1 = 868.62; sensitivity = 0.7802; specificity = 0.7784.

**Figure 3 diagnostics-11-01074-f003:**
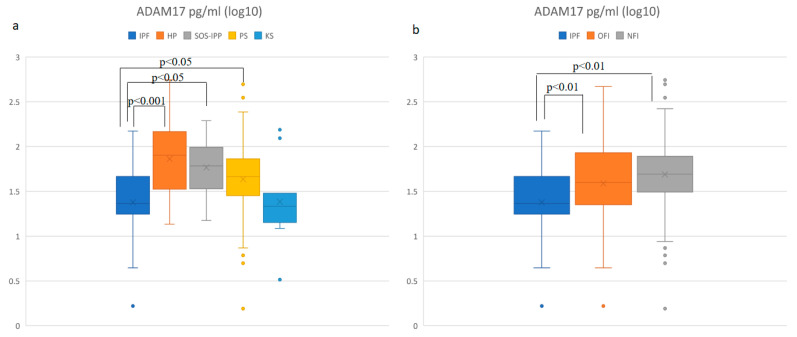
Comparison (log10) of ADAM17 concentration in BALF: (**a**) in different DPLD groups and control group, (**b**) in idiopathic pulmonary fibrosis (IPF), other fibrosis than IPF group (OFI) and non-fibrosis group (NFI). Idiopathic pulmonary fibrosis (IPF)—median: 23.2 pg/mL, interquartile range (IQR): 28.6; hypersensitivity pneumonitis (HP)—median: 82.0 pg/mL, IQR: 120.3; connective tissue disease-associated interstitial lung disease (CTD-ILD)—median: 61.3 pg/mL, IQR: 91.5; pulmonary sarcoidosis (PS)—median: 46.0 pg/mL, IQR: 44.4; control group (CG)—median: 21.6 pg/mL, IQR: 15.9; OFI—median: 48.3 pg/mL, IQR: 72.6; NFI—median: 49.3 pg/mL, IQR: 47.0.

**Figure 4 diagnostics-11-01074-f004:**
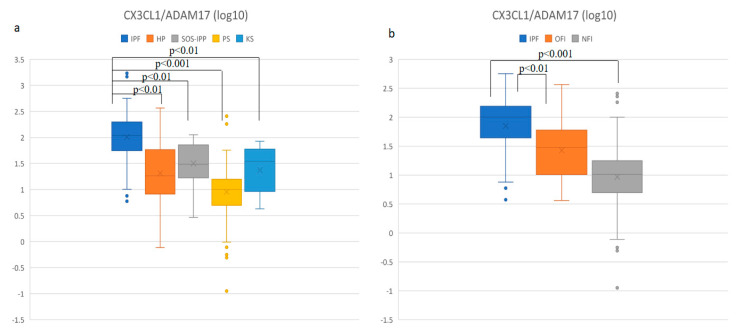
Comparison (log10) of CX3CL1/ADAM17 ratio in BALF: (**a**) in different DPLD groups and control group, (**b**) in idiopathic pulmonary fibrosis (IPF), other fibrosis than IPF group (OFI) and non-fibrosis group (NFI). Idiopathic pulmonary fibrosis (IPF)—median: 105.5, interquartile range (IQR): 72.3; hypersensitivity pneumonitis (HP)—median: 18.2, IQR: 50.0; connective tissue disease-associated interstitial lung disease (CTD-ILD)—median: 26.4, IQR: 42.5; pulmonary sarcoidosis (PS)—median: 10.1, IQR: 10.8; control group (CG) median: 34.6, IQR: 50.4; OFI—median: 30.0, IQR: 72.6; NFI—median: 10.2, IQR: 12.7.

**Figure 5 diagnostics-11-01074-f005:**
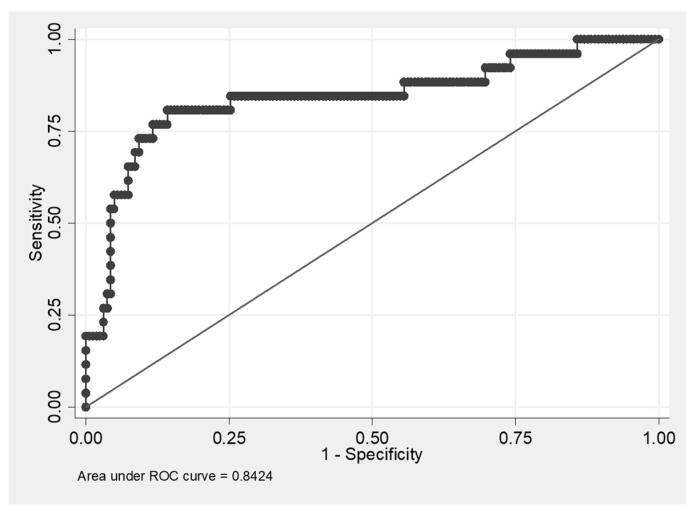
ROC curve of CX3CL1/ADAM17 predictive ability to discriminate IPF (idiopathic pulmonary fibrosis) from other diffuse parenchymal lung diseases. Youden’s index = 0.6657; the cut off value of CX3CL1/ADAM17 ratio associated with the highest value of Youden’s index is CX3CL1/ ADAM17 =41.668; sensitivity = 0.8077; specificity = 0.8580.

**Figure 6 diagnostics-11-01074-f006:**
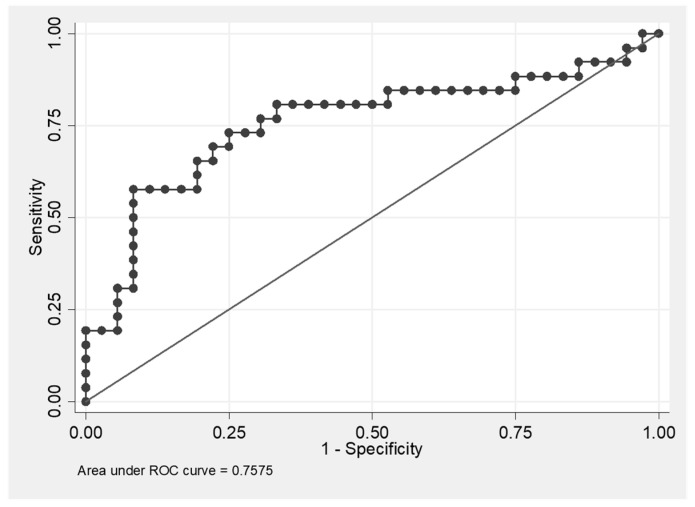
ROC curve of CX3CL1/ADAM17 predictive ability to discriminate IPF (idiopathic pulmonary fibrosis from OFI (other fibrosis than IPF). Youden’s index = 0.4936; the cut off value of CX3CL1/ADAM17 ratio associated with the highest value of Youden’s index is CX3CL1/ ADAM17 = 99.092; sensitivity = 0.5769; specificity = 0.9167.

**Table 1 diagnostics-11-01074-t001:** Characteristics of the study groups.

	PS	IPF	HP	CTD-ILD	CG
Subjects *	156	46	46	22	16
Age (mean)	46	69	49	63	49
Sex (Male/Female) *	90/66	30/16	30/16	7/15	12/4
Smoking status * (current/former/never)	20/27/109	10/22/14	3/10/33	4/7/11	3/4/9
DLCO (% pred) †	87 (15)	53 (12)	62 (16)	58 (15)	94 (21)
Fibrotic stage (Yes/No) *	7/149	46/0	21/25	18/4	0/16
BALF—cell count/μL ^‡^	120 (84)	102 (82)	319 (238)	110 (122)	79 (67)
BALF—Mac (%) ^‡^	58 (32)	75 (18)	28 (25)	61 (31)	87 (9)
BALF—Neu (%) ^‡^	3 (6)	10 (17)	5 (7)	6 (11)	4 (8)
BALF—Ly (%) ^‡^	36 (31)	7 (6)	61 (34)	21 (20)	7 (6)
BALF—CD3 (%) ^‡^	93 (4)	87 (9)	92 (6)	88 (11)	81 (10)
BALF—CD4 (%) ^‡^	72 (21)	49 (23)	42 (37)	43 (36)	49 (19)
BALF—CD8 (%) ^‡^	19 (19)	36 (22)	43 (41)	42 (37)	33 (12)
BALF—CD4/CD8 ^‡^	3.7 (5.0)	1.3 (1.5)	1.0 (4.6)	1.0 (1.7)	1.3 (1.1)

* data presented as n, † data presented as mean (standard deviation), ‡ data presented as median (interquartile range). PS—pulmonary sarcoidosis, IPF—idiopathic pulmonary fibrosis, HP—hypersensitivity pneumonitis, CTD-ILD—connective tissue disease-associated interstitial lung disease, CG—control group, HRCT—high-resolution computed tomography, DLCO—diffusion lung capacity for carbon monoxide, % pred—percentage of predicted value, BALF—bronchoalveolar lavage fluid, Mac—macrophages, Neu—neutrophils, Ly—lymphocytes, CD3—T-cells, CD4—T-helper cells, CD8—cytotoxic T-cells.

**Table 2 diagnostics-11-01074-t002:** Logit models of CX3CL1 effect on fibrotic process.

	Model (1)	Model (2)
Fibrotic process (IPF + OFI)		
CX3CL1	0.000762 ***	0.000586 ***
	(*p* = 0.000)	(*p* = 0.000)
Age		0.120 ***
		(*p* = 0.000)
Sex		0.694
		(*p* = 0.083)
Ex-smoker		0.221
		(*p* = 0.617)
Current smoker		0.490
		(*p* = 0.404)
*N*	285	273
pseudo *R*^2^	0.179	0.404

*p*-values in parentheses, *** *p* < 0.001; IPF—idiopathic pulmonary fibrosis, OFI—other fibrosis than IPF; Model (1) represents logit model of CX3CL1 effect on binary variable fibrotic process. Model (2) represents logit model of CX3CL1 effect on binary variable fibrotic process after controlling effects of other factors (age, sex and smoking).

**Table 3 diagnostics-11-01074-t003:** Correlation analysis of CX3CL1 with BALF cell count and DLCO (all study groups).

CX3CL1 (pg/mL)		
	r *	*p* Value
DLCO % pred	−0.45475	<0.0001
BALF—cell count/μL	0.20252	0.0003
BALF—Mac (%)	−0.05081	0.3718
BALF—Mac (abs)	0.17195	0.0023
BALF—Neu (%)	0.27287	<0.0001
BALF—Neu (abs)	0.35674	<0.0001
BALF—Ly (%)	−0.13920	0.0140
BALF—Ly (abs)	−0.01664	0.7704
BALF—CD3 (%)	−0.14968	0.0082
BALF—CD3 (abs)	−0.01713	0.7635
BALF—CD4 (%)	−0.35272	<0.0001
BALF—CD4 (abs)	−0.11397	0.0446
BALF—CD8 (%)	0.34571	<0.0001
BALF—CD8 (abs)	0.14061	0.0131
BALF—CD4/CD8	−0.35010	<0.0001

* Spearman correlation, DLCO—diffusion lung capacity for carbon monoxide, % pred—percentage of predicted value, BALF—bronchoalveolar lavage fluid, abs—absolute count, Mac—macrophages, Neu—neutrophils, Ly—lymphocytes, CD3—T-cells, CD4—T-helper cells, CD8—cytotoxic T-cells.

**Table 4 diagnostics-11-01074-t004:** Correlation analysis of ADAM17 with cell count in BALF and with DLCO (all study groups).

ADAM17 (pg/mL)		
	r *	*p* Value
DLCO % pred	−0.00945	0.8904
BALF—cell count/μL	0.49927	<0.0001
BALF—Mac (%)	−0.29936	<0.0001
BALF—Mac (abs)	0.29944	<0.0001
BALF—Neu (%)	−0.14157	0.0354
BALF—Neu (abs)	0.13762	0.0405
BALF—Ly (%)	0.39393	<0.0001
BALF—Ly (abs)	0.51110	<0.0001
BALF—CD3 (%)	0.23604	0.0004
BALF—CD3 (abs)	0.51137	<0.001
BALF—CD4 (%)	0.01233	0.8550
BALF—CD4 (abs)	0.47316	<0.0001
BALF—CD8 (%)	0.02179	0.7468
BALF—CD8 (abs)	0.48314	<0.0001
BALF—CD4/CD8	−0.01211	0.8576

* Spearman correlation, DLCO—diffusion lung capacity for carbon monoxide, % pred—percentage of predicted value, BALF—bronchoalveolar lavage fluid, abs—absolute count, Mac—macrophages, Neu—neutrophils, Ly—lymphocytes, CD3—T cells, CD4—T-helper cells, CD8—cytotoxic T cells.

**Table 5 diagnostics-11-01074-t005:** Logit models of CX3CL1/ADAM17 effect on IPF.

	Model (1)	Model (2)
IPF		
CX3CL1/ADAM17	0.0119 ***	0.0109 *
	(*p* = 0.000)	(*p* = 0.017)
Age		0.220 ***
		(*p* = 0.000)
Sex		0.690
		(*p* = 0.419)
Ex-smoker		1.826 *
		(*p* = 0.045)
Current smoker		5.141 **
		(*p* = 0.002)
*N*	199	191
pseudo *R*^2^	0.218	0.635

*p*-values in parentheses, * *p* < 0.05, ** *p* < 0.01, *** *p* < 0.001; IPF—idiopathic pulmonary fibrosis; Model (1) represents logit model of CX3CL1/ADAM17 effect on binary variable IPF. Model (2) represents logit model of CX3CL1/ADAM17 effect on binary variable IPF after controlling effects of other factors (age, sex and smoking).

## Data Availability

The data presented in this study are available on request from the corresponding author.

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
