# Peer review of "The Role of CX3CL1 and ADAM17 in Pathogenesis of Diffuse Parenchymal Lung Diseases"

_diagnostics, 2021, doi:10.3390/diagnostics11061074_

Round 1
Reviewer 1 Report
Please refer to the uploaded file.

Author Response
Dear reviewer, thank you for your revision of our manuscript. We highly appreciate your opinion and review of the paper. We would like to answer your questions and give you more additional information about the study.
The role of CX3CL1 and ADAM17 in pathogenesis of diffuse parenchymal lung diseases Author comments Urban et al. studied the role of CX3CL1 and ADAM17 in the pathogenesis of DPLD by evaluating the level of CX3CL1, and the ratio of CX3CL1 and ADAM17 in the BAL fluid of different DPLD patients including IPF, HP, pulmonary sarcoidosis, and CTD-ILD compared to the normal control. I appreciate the tremendous effort of the authors in carrying out this critical study. While biomarkers for DPLD provide valuable diagnostic tools and are sorely needed, the current study has significant limitations.
Major concerns:
- The clinical context of the subjects has a significant impact on their BAL cellular profiles. The inclusion criteria were a bit too simplistic to capture this issue. At the minimum, it would be very helpful to have 1 table describing demographic, stage of diagnosis (acute exacerbation, stable disease), diagnosis, and treatment at the time of BAL. It would be terrific if all of the BAL procedures were performed without treatment.
Response:
In table 5, we are describing our study group and stating basic demographic parameters (age, gender, and smoking status). Related to diagnosis and treatment, all patients included in our study were newly diagnosed with stable disease. All patients out of the IPF, HP, and PS study groups were at the time of BALF sampling without any antifibrotic and immunosuppressive agents. From these groups were excluded all patients who had already been on this treatment. In our hospital, BAL is a standard procedure during the diagnostic process of DPLDs. We usually don´t perform BAL for the purpose of follow up the activity of treated nor exacerbated DPLDs. Therefore, all patients diagnosed with primary pulmonary DPLDs (IPF, HP, and PS) included in our study were stable and treatment-naive. Only patients from the CTD-ILD group were treated by immunosuppressants at the time of establishing the diagnosis of CTD-ILD. This medication had been previously prescribed by a rheumatologist. Based on our results, there was found no statistical difference of CX3CL1 level between treatment-naive diagnoses (IPF, HP) and CTD-ILD (on immunosuppressive medications). We conclude that immunosuppressive treatment has no impact on the CX3CL1 level.
We suggest adding more information about diagnosis and treatment to the text as follows:
Into the part Materials and Methods, behind the sentence: ´All subjects included in the study had no other respiratory comorbidities and no history of active malignant disease.´ All patients included in our study were newly diagnosed with stable disease. All patients out of the IPF, HP, and PS study groups were at the time of BALF sampling without any antifibrotic and immunosuppressive treatment. Patients from the CTD-ILD group were treated by immunosuppressive agents at the time of establishing the diagnosis of CTD-ILD. This medication had been previously prescribed by a rheumatologist.
- I am not sure that normal control is appropriate in this setting. May it be more informative to use non-fibrotic disease as a control?
Response:
Our study is based on the findings of the potential role of CX3CL1 in the pathogenesis of bleomycin-induced pulmonary fibrosis in murine models. At the beginning of our study (2017), we didn´t find any published study focusing on CX3CL1 in humans. Therefore, we decided to create a control group in order to compare CX3CL1 levels within DPLD groups. CG was created from the patients with mediastinal lymphadenopathy (suspicion of I. stage of sarcoidosis) who underwent bronchoscopy with BAL, but after completing all the diagnostic procedures (including biopsy), the DPLD diagnosis was excluded. We also compared the fibrotic and non-fibrotic groups created from our DPLD groups to compare the levels of CX3CL1 in the fibrotic and non-fibrotic DPLDs (IPF vs. OFI vs. NFI).
- Figure 1 showed no difference of CX3CL1 level in BAL between IPF and OFI; why the authors believe that CX3CL1 was significantly higher in IPF? Please indicate statistics and clarify.
Response:
Figure 1 really shows no difference in CX3CL1 levels between IPF and OFI. A significant difference was found between both fibrotic groups and non-fibrotic group - IPF vs. NFI and OFI vs. NFI as well. Based on these results, we stated that CX3CL1 has an important role in the pathogenesis of pulmonary fibrosis regardless of the etiology (primary/idiopathic or secondary).
- Figure 4, where the level of CX3CL1/ADAM17 statistically different between groups? It seems that the difference of CX3CL1/ADAM17 is driven by the low level of ADAM17, not necessarily due to an absolute high of CX3CL1 per se; please clarify.
Response:
Statistical difference of the CX3CL1/ADAM17 values was found only in the IPF group compared to other DPLD groups. No other statistical differences among other DPLD groups were found. We agree, that the differences of CX3CL1/ADAM17 value could be driven especially by the level of ADAM17. By using the CX3LC1/ADAM17 and its significantly high value in IPF, we wanted to point to a probable different mechanism of cleavage of CX3CL1 in IPF compared to other fibrosing DPLDs.
- What is the cut-off value of CX3CL1 and CX3CL1/ADAM17 to be applied in each ROC?
Response:
Cut-off values were associated with the highest value of Youden's index (point on the ROC, where is the highest value calculated from the formula: sensitivity + specificity – 1).
Figure 2:
The cut-off value of CX3CL1 associated with the highest value of Youden's index is CXCL1=868.62.
Figure 5:
The cut-off value of CX3CL1/ADAM17 ratio associated with the highest value of Youden's index is CX3CL1/ ADAM17 =41.668.
Figure 6:
The cut-off value of CX3CL1/ADAM17 ratio associated with the highest value of Youden's index is CX3CL1/ ADAM17 =99.092.
We suggest adding information about the cut-off value to figure legends of each chart as follows:
Figure 2. ROC curve of CX3CL1 predictive ability to discriminate fibrotic diffuse parenchymal lung disease (DPLD) from other (non-fibrotic) DPLD. Youden's index = 0.5586; the cut-off value of CX3CL1 associated with the highest value of Youden's index is CX3CL1=868.62; Sensitivity = 0.7802; Specificity = 0.7784.
Figure 5. ROC curve of CX3CL1/ADAM17 predictive ability to discriminate IPF (idiopathic pulmonary fibrosis) from other diffuse parenchymal lung diseases. Youden's index = 0.6657; the cut-off value of CX3CL1/ADAM17 ratio associated with the highest value of Youden's index is CX3CL1/ ADAM17 =41.668; Sensitivity = 0.8077; Specificity = 0.8580.
Figure 6. The ROC curve of CX3CL1/ADAM17 predictive ability to discriminate IPF (idiopathic pulmonary fibrosis from OFI (other fibrosis than IPF). Youden's index = 0.4936; the cut-off value of CX3CL1/ADAM17 ratio associated with the highest value of Youden's index is CX3CL1/ ADAM17 =99.092; Sensitivity = 0.5769; Specificity = 0.9167
- It seems that the data suggested a correlative effect of CX3CL1 on IPF rather than a causal effect on IPF pathogenesis. More studies are required, such as inhibiting CX3CL1 to implicate CX3CL1 role. Please elaborate on this issue.
Response:
We agree, that at present it is not possible to proclaim if CX3CL1 causes a fibrotic process (not only IPF) or is a consequence of other mechanisms acting in the pathogenesis of the fibrotic process.
However, this study was not focused on clarifying molecular mechanisms underlying the fibrotic process and the main goal of the study was BAL fluid detection of molecules, identified as important players in fibrosis by other study groups. The study proved that the molecules can be identified in BAL fluid in humans and showed the possibility of using the measurement in clinical practice as markers in the process of differential diagnosis.
The results open new possibilities and interesting hypotheses, we plan to work on in our further research. The extent of this manuscript has already oversized the limit required by most journals.
There is a need for the following research. We intend to continue in our study by investigating new molecules which could play a role in the pathogenesis of the fibrotic process. However, our results could be an inspiration for other research teams to confirm or augment these findings.
We suggest adding one sentence in the Discussion part, behind the sentence: ´In line with our results, we assume that the higher CX3CL1 concentration in the lung parenchyma intensifies the attraction of cytotoxic cells, including CD8+ T cells, enhancing the fibrotic milieu. The inflammation-damaged bronchoepithelial cells are assumed abundant source of CX3CL1 (21).´ Nowadays, it is not clear if CX3CL1 is a causal factor in the fibrotic process (not only in IPF) or only the consequence of other mechanisms acting in the pathogenesis of the fibrotic process. There is a need for continuing research.
Minor concerns:
- Can the authors explain the utility of using log 10 scaling for comparison?
Response:
The purpose of using log 10 scaling for comparison was a better visualization of presenting results in graphs. By using the standard comparison were charts depressed to the abscissa axis due to some extreme values.
- If ADAM17 cleaves CX3CL1, why a higher level of CX3CL1 is observed in IPF despite a low level of ADAM17.
Response:
On the basis of our study, it is not possible to explain the mechanism responsible for releasing the soluble CX3CL1 in IPF. There are surely other mechanisms playing the role in the shedding of CX3CL1 in IPF (ADAM10, or other). Further studies are needed to elucidate the pathomechanism.

Reviewer 2 Report
The oaoer entitle: “The role of CX3CL1 and ADAM17 in pathogenesis of diffuse 2 parenchymal lung diseases” by Jan Urban et colleagues resulted novelty and of interest in the field of DILD.
However some point can be improved:
- Introduction resulted well written and clear, I only suggest to better clarify the aims of the study
- I also suggest to shorten the introduction about ILD, the readers interested in this paper know the problem of prognosis and fibrotic progression.
- RESULTS: the sentences “Based on the results, it can be stated: First, the elevated levels of CX3CL1 in BALF corre- 112 lates highly significantly with the development of fibrosis. Second, higher levels of 113 CX3CL1 are associated with an increased probability of the fibrotic lung process. CX3CL1 114 data remained highly significant even after controlling of others specific parameters (age, 115 gender and smoking).” I think that resulted inappropriate in this section. I suggest to move this part in discussion followed by a comment.
- Regarding the section : “Positive correlation of the level of CX3CL1 with number of macrophages, neutrophils and 135 CD8+ T cells, and a negative correlation with the number of CD4+ T cells and with DLCO values” I suggest to the author to describe only very strong correlation (r≥±0.6) and remove the table (or adding as supplementary materials.
- The same it is true for the section: “Positive correlation of the level of ADAM17 with absolute cell count in BALF”
- Regarding materials and methods section I suggest to divide the parts in order to simplify the readers.
- I also suggest to add the gating strategy that was used for lymphocytes immunophenotyping through flow cytometry as figure or supplementary figures.
Author Response
Dear reviewer, thank you for your revision of our manuscript. We highly appreciate your opinion and review of the paper. We would like to react to your suggestions.
The oaoer entitle: “The role of CX3CL1 and ADAM17 in pathogenesis of diffuse 2 parenchymal lung diseases” by Jan Urban et colleagues resulted novelty and of interest in the field of DILD.
However some point can be improved:
- Introduction resulted well written and clear, I only suggest to better clarify the aims of the study
Response: We tried to specify all aims in the best possible way. I think we mentioned all the important aims of the study. Nevertheless, we could emphasize that the study focused on the evaluation of the role of CX3CL1 in the pathogenesis of the fibrotic process and the potential role of explored molecules to distinguish fibrotic and non-fibrotic processes. We suggest reformulating the part with stated aims:
In this study, we evaluated the potential role of CX3CL1 in the pathogenesis of the fibrotic process of various DPLDs. The aim of this study was to determine the level of CX3CL1 in bronchoalveolar lavage fluid (BALF) of patients affected by particular DPLDs – pulmonary sarcoidosis (PS), idiopathic pulmonary fibrosis (IPF), hypersensitivity pneumonitis (HP), and connective tissue disease-associated interstitial lung disease (CTD-ILD). We compared concentrations of CX3CL1 in each of these populations, in particular fibrotic and non-fibrotic stages of DPLDs. Next, we correlated the level of CX3CL1 with the number of cells in BALF and the severity of lung impairment evaluated by measurement of lung diffusion capacity for carbon monoxide (DLCO). In the light of previously published data regarding ADAM17 (10, 11) acting as an enzyme releasing soluble form of CX3CL1, we also evaluated the relationship of CX3CL1 to ADAM17 concentrations in BALF. Finally, we statistically evaluated the ability to distinguish primary fibrotic process (IPF), from secondary fibrotic (fibrotic stages of HP, PS, CTD-ILD), and non-fibrotic DPLDs on the basis of the levels of studied molecules in BALF.
- I also suggest to shorten the introduction about ILD, the readers interested in this paper know the problem of prognosis and fibrotic progression.
Response: We fully agree with your suggestion to shorten the introduction part generally describing ILD. We suggest reformulating this part:
Diffuse parenchymal lung diseases (DPLDs) comprise a broad heterogeneous group of diseases with more than 200 various diagnostic units. The common feature of DPLDs is the involvement of lung interstitium characterized by different degree of inflammation and fibrosis with extracellular matrix accumulation (1). The prognosis depends on the underlying immunopathological process. The continuing inflammatory process can result in the activation of the fibrosing process (2). The worst prognosis is related to the progressive fibrotic phenotype of disease (3). The main problem of DPLDs in regard to the number of diagnostic units is the differential diagnosis and the limited treatment possibilities. The cornerstone to move ahead in therapeutic options is an improvement of comprehension of immunopathogenesis.
- RESULTS: the sentences “Based on the results, it can be stated: First, the elevated levels of CX3CL1 in BALF corre- 112 lates highly significantly with the development of fibrosis. Second, higher levels of 113 CX3CL1 are associated with an increased probability of the fibrotic lung process. CX3CL1 114 data remained highly significant even after controlling of others specific parameters (age, 115 gender and smoking).” I think that resulted inappropriate in this section. I suggest to move this part in discussion followed by a comment.
Response: We agree that this explanation would be more appropriate for discussion but we wanted to comment shortly on the results presented in table 1. We prefer not to move all these notes to the discussion but only to reformulate and shorten this part to:
.....we adopted logistic regression models (Table 1). The elevated levels of CX3CL1 in BALF highly significantly correlate with the development of fibrosis and raised levels of CX3CL1 are associated with an increased probability of the fibrotic lung process. CX3CL1 data remained highly significant even after controlling other specific parameters (age, gender, and smoking).
- Regarding the section : “Positive correlation of the level of CX3CL1 with number of macrophages, neutrophils and 135 CD8+ T cells, and a negative correlation with the number of CD4+ T cells and with DLCO values” I suggest to the author to describe only very strong correlation (r≥±0.6) and remove the table (or adding as supplementary materials.
Response: The calculated p-values shown in Table 3 support the high correlative probability of these molecules with subtypes of the immune cells in BALF and DLCO despite lower r-values and we regarded it as interesting to mention at least correlations with p< 0.0001 in the text and leave in the correlation table close to the sentences reporting this correlation (according to the instructions of the editor). However, we suggest refining the statement about strong correlation by leaving out the words ´strong´ or ´strongly´ in the following sentences: ´Correlation analysis of CX3CL1 concentration with a differential cell count in BALF in all study groups pointed to a strong positive correlation with an absolute BALF´s cell count, absolute count of neutrophils, macrophages and CD8+ (cytotoxic) T cells, and a strong negative correlation with an absolute count of CD4+ (helper) T cells and with CD4+/CD8+ T cells ratio. The CX3CL1 concentration moreover strongly inversely correlated with DLCO levels (Table 3).´
And reword this sentence to: Correlation analysis of CX3CL1 concentration with a differential cell count in BALF in all study groups pointed to a positive correlation with absolute BALF´s cell count, absolute count of neutrophils, macrophages, CD8+ (cytotoxic) T cells, and a negative correlation with an absolute count of CD4+ (helper) T cells, CD4+/CD8+ T cells ratio. Moreover, the CX3CL1 concentration inversely correlated with DLCO levels (Table 3).
- The same it is true for the section: “Positive correlation of the level of ADAM17 with absolute cell count in BALF”
Response: The calculated p-values in Table 3 support the high correlative probability of these molecules with subtypes of the immune cells in BALF and DLCO despite lower r-values and we regarded it interesting to mention at least correlations with p< 0.0001 in the text and leave in the correlation table close to the sentences reporting these correlations (according to the instructions of the editor). Nevertheless, we suggest refining the statement about strong correlation by leaving out the word ´strong´ in the sentence: ´Correlation analysis of ADAM17 concentration with differential cell counts in BALF in all study groups pointed to a strong positive correlation with the absolute BALF´s cell count, absolute count of lymphocytes (both CD4+ T cells and CD8+ T cells) and macrophages (Table 4).´
And reword this sentence to: Correlation analysis of ADAM17 concentration with differential cell counts in BALF in all study groups pointed to a positive correlation with the absolute BALF´s cell count, absolute count of lymphocytes (both CD4+ T cells and CD8+ T cells) and macrophages (Table 4).
- Regarding materials and methods section I suggest to divide the parts in order to simplify the readers.
Respond: Thank you for your suggestion. We agree to divide the methods section into three parts to simplify to orientation in this pars for readers.
- Materials and methods
4.1. Study group
The study group comprised 270 DPLD patients … All subjects included in the study had no other respiratory comorbidities and no history of active malignant disease.
4.2. Procedures and sample processing
BALF samples were obtained ...using the single breath method in accordance with “2017 ATS/ERS standards” (37).
4.3. Statistical analysis
The one-sample Kolmogorov-Smirnov test ……... Statistical analysis was performed using the SAS and Stata software.
- I also suggest to add the gating strategy that was used for lymphocytes immunophenotyping through flow cytometry as figure or supplementary figures.
Response: Thank you for your suggestion. Considering the aims of the study focusing on new molecules in DLPDs measured by ELISA method we think that figures depicting lymphocytes immunophenotyping strategy by flow cytometry would be out of interest to readers. But we could provide this in Appendix as Figure A1:
Appendix A
Figure A1. Immunophenotyping of lymphocytes

Round 2
Reviewer 1 Report
I appreciated the authors’ effort in revising their manuscript and providing point-by-point response.
Additional suggestion:
1) In all figures and graphs, please add asterisks (*) or (**), or (***) according to the level of significance statistically.
2) I would consider adding a few limitations of this study, which include a small sample size and the lack of validation of the propose biomarkers.
Reviewer 2 Report
now the paper can be published